DOI: 10.1038/s41467-018-04448-6　　**OPEN**

# The genomic landscape of *TERT* promoter wildtype-*IDH* wildtype glioblastoma

Bill H. Diplas [1,2], Xujun He [1,2,3], Jacqueline A. Brosnan-Cashman [4], Heng Liu [1,2], Lee H. Chen [1,2], Zhaohui Wang [1,2], Casey J. Moure [1,2], Patrick J. Killela [1,2], Daniel B. Loriaux [1,2], Eric S. Lipp [1], Paula K. Greer [1,2], Rui Yang [1,2], Anthony J. Rizzo [4], Fausto J. Rodriguez [4], Allan H. Friedman [1,5], Henry S. Friedman [1], Sizhen Wang [6], Yiping He [1,2], Roger E. McLendon [1,2], Darell D. Bigner [1,5], Yuchen Jiao [7], Matthew S. Waitkus [1,2], Alan K. Meeker [4] & Hai Yan [1,2]

The majority of glioblastomas can be classified into molecular subgroups based on mutations in the *TERT* promoter (*TERTp*) and isocitrate dehydrogenase 1 or 2 (*IDH*). These molecular subgroups utilize distinct genetic mechanisms of telomere maintenance, either *TERTp* mutation leading to telomerase activation or *ATRX*-mutation leading to an alternative lengthening of telomeres phenotype (ALT). However, about 20% of glioblastomas lack alterations in *TERTp* and *IDH*. These tumors, designated *TERTp*[WT]-*IDH*[WT] glioblastomas, do not have well-established genetic biomarkers or defined mechanisms of telomere maintenance. Here we report the genetic landscape of *TERTp*[WT]-*IDH*[WT] glioblastoma and identify *SMARCAL1* inactivating mutations as a novel genetic mechanism of ALT. Furthermore, we identify a novel mechanism of telomerase activation in glioblastomas that occurs via chromosomal rearrangements upstream of *TERT*. Collectively, our findings define novel molecular subgroups of glioblastoma, including a telomerase-positive subgroup driven by *TERT*-structural rearrangements (*IDH*[WT]-*TERT*[SV]), and an ALT-positive subgroup (*IDH*[WT]-ALT) with mutations in *ATRX* or *SMARCAL1*.

[1] The Preston Robert Tisch Brain Tumor Center at Duke, Duke University Medical Center, Durham, 27710 NC, USA. [2] Department of Pathology, Duke University Medical Center, Durham, 27710 NC, USA. [3] Key Laboratory of Gastroenterology of Zhejiang Province, Zhejiang Provincial People's Hospital, Hangzhou Medical College, Hangzhou, 310014, China. [4] Department of Pathology, Sidney Kimmel Comprehensive Cancer Center at Johns Hopkins, Johns Hopkins University School of Medicine, Baltimore, 21231 MD, USA. [5] Department of Neurosurgery, Duke University Medical Center, Durham, 27710 NC, USA. [6] Genetron Health (Beijing) Co. Ltd, Beijing, 102208, China. [7] State Key Laboratory of Molecular Oncology, Laboratory of Cell and Molecular Biology, National Cancer Center/Cancer Hospital, Chinese Academy of Medical Sciences and Peking Union Medical College, Beijing, 100021, China. These authors jointly supervised this work: Hai Yan, Alan K. Meeker, Matthew S. Waitkus, Yuchen Jiao. Correspondence and requests for materials should be addressed to Y.J. (email: jiaoyuchen@cicams.ac.cn) or to M.S.W. (email: matthew.waitkus@duke.edu) or to A.K.M. (email: ameeker1@jhmi.edu) or to H.Y. (email: hai.yan@duke.edu)

Glioblastoma (GBM, World Health Organization (WHO) grade IV) is the most common and deadly primary brain tumor with a median overall survival (OS) of less than 15 months despite aggressive treatment[1,2]. There is a critical need for molecular markers for GBM to improve personalized diagnosis and treatment, and for a better understanding of the underlying biology to inform the development of novel therapeutics.

This report presents a comprehensive molecular analysis of ~20% of GBMs that lack established genetic biomarkers or defined mechanisms of telomere maintenance[3]. These are aggressive tumors that are known as $TERTp^{WT}$-$IDH^{WT}$ GBMs, a largely unknown territory as they lack mutations in the most commonly used biomarkers, isocitrate dehydrogenase 1 and 2 (IDH)[4–6] and the promoter region of telomerase reverse transcriptase (TERTp)[5–7].

TERTp and IDH mutations are routinely used clinically to facilitate diagnosis by classifying 80% of GBMs into molecular subgroups with distinct clinical courses[4–13]. Each GBM molecular subgroup also utilizes different mechanisms of telomere maintenance. The TERTp-mutant GBMs exhibit telomerase activation, due to generation of de novo transcription factor binding sites leading to increased TERT expression[5,14–16], while the IDH-mutant GBMs exhibit alternative lengthening of telomeres (ALT) due to concurrent loss-of-function mutations in ATRX[3,10,13,17–20]. Based on these patterns, genetic alterations enabling telomere maintenance are likely to be critical steps in gliomagenesis.

Here, we use whole exome sequencing (WES) and whole genome sequencing (WGS) to define the mutational landscape of $TERTp^{WT}$-$IDH^{WT}$ GBM. We identify recurrently mutated genes and pathways in this tumor subset. Most notably, we identify novel somatic mutations related to mechanisms of telomere maintenance. These include recurrent genomic rearrangements upstream of TERT (50%) leading to increased TERT expression, and alterations in ATRX (21%) or SMARCAL1 (20%) in ALT-positive $TERTp^{WT}$-$IDH^{WT}$ GBMs. We report the discovery of somatic SMARCAL1 loss-of-function mutations and their involvement in ALT-mediated telomere maintenance in cancer. Using a variety of cell-based assays, we show the role of SMARCAL1 as an ALT suppressor and genetic factor involved in telomere maintenance. Finally, we identify an enrichment of several therapeutically targetable alterations in $TERTp^{WT}$-$IDH^{WT}$ GBM, including mutations in BRAF V600E (20%). These findings define the core molecular alterations of this important subset of GBM and identify novel targets for a disease lacking effective therapies.

## Results

### The genetic landscape of $TERTp^{WT}$-$IDH^{WT}$ GBM.
We identified a cohort of patients with tumors that were $TERTp^{WT}$-$IDH^{WT}$ by screening 260 GBMs for mutations in the TERT promoter and IDH1/2. Forty-four $TERTp^{WT}$-$IDH^{WT}$ cases were identified, which comprised 16.9% of the total GBM cohort[4]. The $TERTp^{WT}$-$IDH^{WT}$ GBMs with available 1p/19q status available did not display 1p/19q co-deletion, consistent with previous reports that have labeled these tumors "triple-negative" due to the observation that they lack all three common diffuse glioma biomarkers ($TERTp^{WT}$-$IDH^{WT}$-1p/19q$^{WT}$)[8]. The age distribution of the $TERTp^{WT}$-$IDH^{WT}$ GBM cohort was bimodal, with one mode at 28 years and the other at 56 years (range: 18 to 82 years). Approximately 30% (13/44) of $TERTp^{WT}$-$IDH^{WT}$ GBMs were younger than 40 years old (Fig. 1, Supplementary Figure 1, Supplementary Data 1-2). We performed WES on cases for which DNA from untreated tumor tissue and matched peripheral blood were available (Discovery cohort, $N = 25$). The average sequencing coverage was 140-fold (range: 70 to 265) and 92% of bases

had at least 10 high-quality reads (range: 87 to 94%). We identified 1449 total somatic, non-synonymous mutations in the exomes of the $TERTp^{WT}$-$IDH^{WT}$ GBMs, with each having an average of 58 mutations per tumor (range: 6 to 431, Fig. 1), resulting in an average mutation rate of approximately 1.74 coding mutations per Mb, similar to rates observed in GBMs from previous studies (1.5 mutations/Mb)[7].

The mutational landscape of $TERTp^{WT}$-$IDH^{WT}$ GBM is shown in Fig. 1. Recurrently mutated genes in $TERTp^{WT}$-$IDH^{WT}$ GBM occurred in pathways including the RTK/RAS/PI3K (88%), P53 (40%), and RB (24%) pathways (Fig. 1, Supplementary Data 3-5). Additional genes harboring copy number variations included PDGFRA (8%), MDM2 and MDM4 (12%), CDKN2B (12%), and CDK4 (Fig. 1, Supplementary Data 5). At least one recurrently mutated gene ($n \geq 2$) was identifiable in 92% of the $TERTp^{WT}$-$IDH^{WT}$ GBMs.

IntOGen analysis[21,22] identified several known glioma-associated driver alterations ($P < 0.05$, $n \geq 2$), including PTEN (32%), NF1 (24%), EGFR (28%), TP53 (24%), ATRX (20%), and BRAF (20%), as well as two novel candidate drivers, SMARCAL1 (16%) and PPM1D (8%) (Supplementary Data 6), both of which have not previously been implicated as drivers in adult supratentorial GBM. All mutations identified in the serine/threonine protein kinase BRAF were V600E, the clinically actionable hotspot mutation that causes increased kinase activity and RAS pathway activation. BRAF mutations occurred significantly more often than previous studies (20% vs. 1.7% of GBM[23], $P = 0.0007$, two-sided Fisher's exact test). Most of these alterations (4/5, 80%) were present in adult patients ≤ 30 years old ($P = 0.0019$, two-sided Fisher's exact test). The PPM1D mutations identified were located in the C-terminal regulatory domain (exon 6), leading to a truncated protein with an intact phosphatase domain, similar to PPM1D mutations described in gliomas of the brainstem[11].

### SMARCAL1-mutant GBMs exhibit hallmarks of ALT.
The mutations identified in the novel candidate driver SMARCAL1 were primarily nonsense or frameshift with mutant allele fractions greater than 50% (average: 69%; range: 59–83%), indicating likely loss of heterozygosity and a loss-of-function mutational pattern. SMARCAL1 encodes an adenosine triphosphate (ATP)-dependent annealing helicase that has roles in catalyzing the rewinding of RPA-bound DNA at stalled replication forks[24,25], and was recently shown to be involved in resolving telomere-associated replication stress[26,27]. SMARCAL1 has similarities with ATRX, which is also a member of the SWI/SNF family of chromatin remodelers and has both ATP-binding and C-terminal helicase domains[28]. Additionally, ATRX harbors recurrent loss-of-function mutations that result in loss of nuclear expression in ALT-positive gliomas[10,13,17].

Given these similarities to ATRX, we sought to determine if SMARCAL1-mutant tumors exhibit markers of ALT, including C-circles and ultrabright telomeric foci (telomere fluorescent in situ hybridization (FISH))[20,29]. We expanded the cohort of $TERTp^{WT}$-$IDH^{WT}$ GBMs ($N = 39$) and sequenced SMARCAL1, identifying mutations in 21% (8/39) of tumors, with the majority (75%, 6/8) of these alterations being frameshift, nonsense, or splice site mutations (Fig. 2a). All SMARCAL1-mutant GBMs exhibited both ultrabright telomeric foci and C-circles, suggesting a novel link between somatic SMARCAL1 loss-of-function mutations in cancer and the ALT mechanism of telomere maintenance. Additionally, by assaying ATRX expression by immunohistochemistry (IHC), we found that loss of nuclear ATRX was observed in 22% (8/37) of $TERTp^{WT}$-$IDH^{WT}$ GBMs. Overall, 36% (14/39) of $TERTp^{WT}$-$IDH^{WT}$ GBMs exhibited both

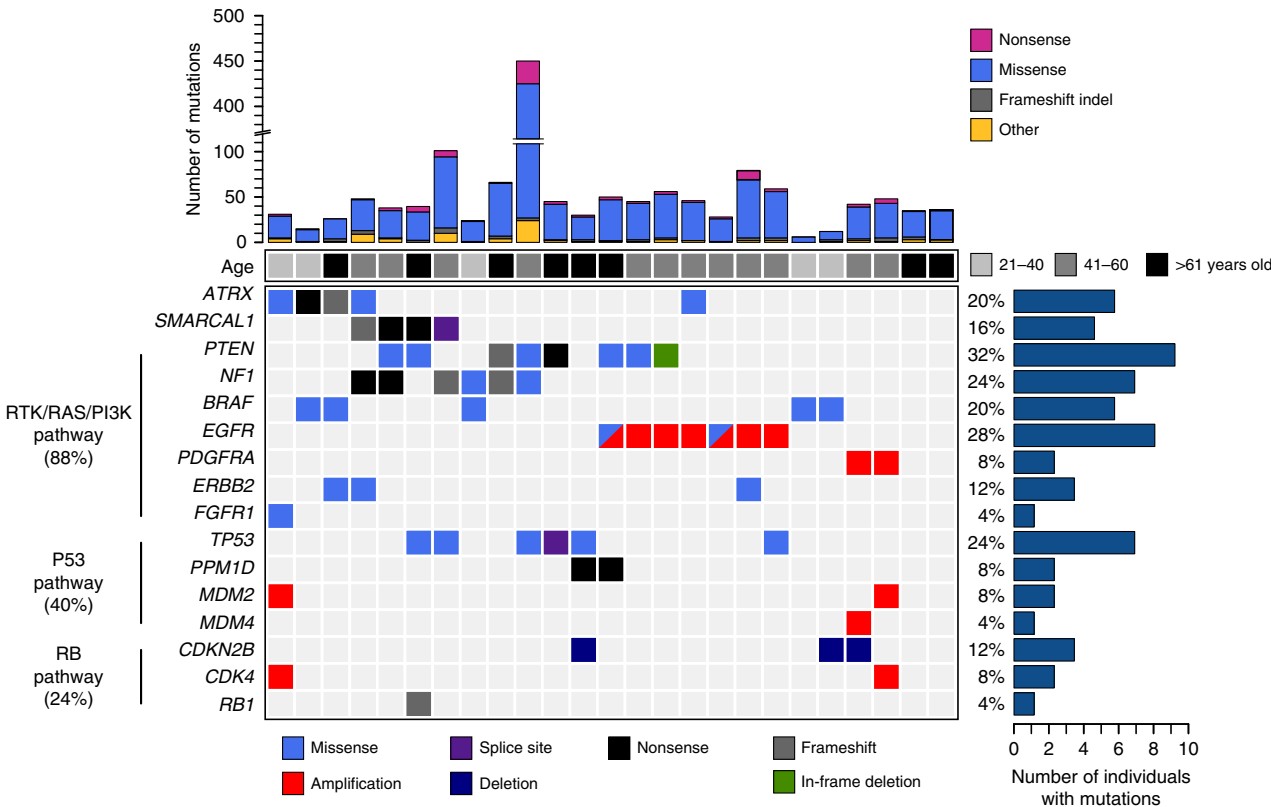

**Fig. 1** The mutational landscape of somatic coding alterations in *TERTp*^WT^-*IDH*^WT^ GBM. Whole exome sequencing was performed on *TERTp*^WT^-*IDH*^WT^ GBMs ($N = 25$). Recurrently mutated pathways identified included the RTK/RAS/PI3K (88%), P53 (40%), and RB (24%) pathways. Somatic mutation rates per case are shown with corresponding patient age (top). Recurrently mutated genes displayed determined to be significantly mutated (IntOgen algorithm, $P < 0.05$, $n \geq 2$) are shown, as well as select lower frequency genes that are recurrently mutated in glioma or known oncogenes/tumor suppressors in the pathways shown. The mutation frequency of each gene is shown (right) as a percentage of the total cohort

ultrabright telomeric foci and C-circles, which are hallmarks consistent with the ALT phenotype. Of these ALT-positive tumors, 46.7% (7/15) showed loss of nuclear *ATRX* expression, while the other 53.3% (8/15) harbored *SMARCAL1* mutations, exhibiting a mutually exclusive pattern ($P = 0.01$, Fisher's exact test, two-tailed, odds ratio = 0.024, Fig. 2a). Finally, based on exome sequencing results, 80% (8/10) of the ALT-positive *TERTp*^WT^-*IDH*^WT^ GBMs also harbored alterations in *NF1* or *BRAF*, indicating a potential molecular signature of co-occurring alterations in RAS-activating and ALT-inducing pathways (Fig. 1).

**Identification of *TERT* rearrangements in *TERTp*^WT^-*IDH*^WT^ GBM.** Based on the measurement of markers of ALT, 61.5% (24/39) of *TERTp*^WT^-*IDH*^WT^ GBMs did not exhibit ultrabright foci or C-circle accumulation (ALT negative), suggesting that these cases may utilize a telomerase-dependent mechanism of telomere maintenance, independent of *TERTp* mutation (Fig. 2a). We sought to identify genetic alterations impacting telomerase activity that would not be detectable by exome sequencing.

We performed WGS on ALT-negative *TERTp*^WT^-*IDH*^WT^ GBMs ($N = 8$) and their paired matched normal genomic DNA (Supplementary Data 7–10). Structural variant analysis[30] identified recurrent rearrangements upstream of *TERT* in 75% (6/8) of the ALT-negative *TERTp*^WT^-*IDH*^WT^ GBMs sequenced (Fig. 2b, c). Half of these rearrangements were translocations to other chromosomes, while the remaining were intrachromosomal inversions. Breakpoints were validated as tumor specific by junction-spanning PCR in five of six cases (Supplementary Figure 2). To detect *TERT* structural variants in the entire

*TERTp*^WT^-*IDH*^WT^ GBM cohort, we used break-apart FISH with probes spanning *TERT* (Fig. 2d, Supplementary Figure 3A, B). In total, we found 50% (19/38) of the *TERTp*^WT^-*IDH*^WT^ GBMs harbored *TERT* structural rearrangements. *TERT*-rearranged GBMs exhibited mutual exclusivity with the ALT-positive *TERTp*^WT^-*IDH*^WT^ GBMs ($P = 0.0019$, Fisher's exact test, two-tailed, odds ratio = 0.069). Analysis of *TERT* messenger RNA (mRNA) expression revealed that *TERT*-rearranged GBMs express significantly higher levels of *TERT* compared to the ALT-positive (*ATRX* and *SMARCAL1*-mutant) *TERTp*^WT^-*IDH*^WT^ GBMs ($P = 0.016$, Kruskal–Wallis test using Dunn's test post hoc, Fig. 2e). This is a similar pattern to that observed between the other two major GBM subtypes, where telomerase-positive, *IDH*^WT^-*TERTp*^MUT^ GBMs exhibit significantly higher *TERT* mRNA expression ($P = 0.0036$, Kruskal–Wallis test using Dunn's test post hoc) relative to the *IDH*^MUT^-*TERTp*^WT^ GBMs, which are *ATRX* mutated and exhibit ALT[10]. There were no significant differences in *TERT* expression between the *TERT*^SV^ and *TERTp* mutant subgroups (or between the *IDH*-mutant and *IDH*^WT^-ALT subgroups). Of the seven remaining ALT-negative tumors that lacked *TERT* rearrangement, one tumor harbored amplification of *MYC*, a known transcriptional activator of *TERT*[31], and this tumor displayed elevated *TERT* expression (Fig. 2e, arrow).

**Telomere-related alterations define new subgroups of GBM.** Using whole exome and genome sequencing, we identified frequent telomere maintenance-related alterations that define new genetic subgroups of GBM. The *IDH*^WT^-ALT GBM subgroup, which harbors *ATRX* and *SMARCAL1* mutations, accounts for

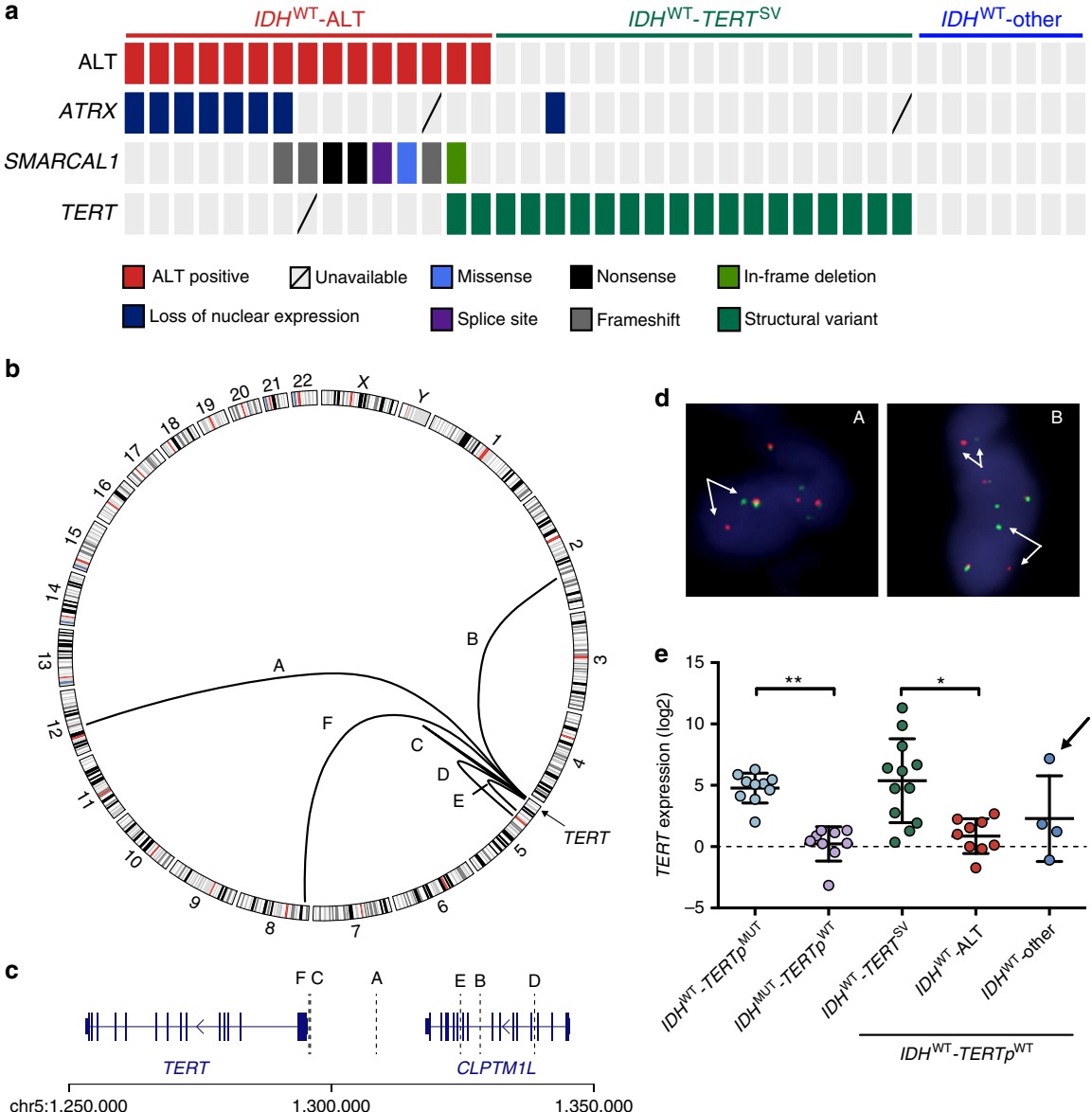

**Fig. 2** Inactivating mutations in *SMARCAL1* and *ATRX*, and rearrangements upstream of *TERT* are frequent in *TERTp*$^{WT}$-*IDH*$^{WT}$ GBMs and related to distinct telomere maintenance mechanisms. **a** Based on ALT assessment by both telomere FISH and C-circle (dot blot), 38.5% (15/39) of *TERTp*$^{WT}$-*IDH*$^{WT}$ GBMs exhibit signs of ALT. Of these, approximately half exhibit loss of ATRX expression (IHC) and half harbor mutations in *SMARCAL1*, in a largely mutually exclusive manner. *TERT* rearrangements were identified by whole genome sequencing ($N = 8$). Break-apart FISH was used to screen the cohort for *TERT* rearrangements, which were present in 50% (19/38) of all *TERTp*$^{WT}$-*IDH*$^{WT}$ GBMs. **b** Circos plot of rearrangements identified upstream of *TERT* by whole genome sequencing of ALT-negative GBMs ($N = 8$). Several cases were interchromosomal translocations (A, B, F), while the remaining cases were intrachromosomal (C, D, E). **c** The breakpoints of the rearrangements identified by whole genome sequencing span a region in the 50 kb upstream of *TERT*. **d** Examples of FISH on patient tumor tissue showing break-apart signal, indicating *TERT*-rearrangement. Arrows identify break-apart signals. **e** *TERT* expression was assessed by rt-qPCR relative to *GAPDH*. *IDH*$^{WT}$-*TERT*$^{SV}$ ($n = 12$) tumors exhibit significantly higher *TERT* expression than the *IDH*$^{WT}$-ALT subgroup ($n = 9$, $P < 0.05$). This is a similar trend seen among known GBM groups, where the *IDH*$^{WT}$-*TERTp*$^{MUT}$ GBMs (telomerase positive) exhibit increased *TERT* expression compared to *IDH*$^{MUT}$-*TERTp*$^{WT}$ (ALT positive) GBMs ($P < 0.01$). The *IDH*$^{WT}$-other subgroup is ALT negative, but does not harbor detectable *TERT* rearrangements. One case in this group harbors *MYC* amplification (arrow), known to increase *TERT* expression due to the presence of *MYC* binding sites in the *TERT* promoter region. Error bars in e denote s.d. *$P < 0.05$; **$P < 0.01$; Kruskal–Wallis test with Dunn's multiple comparisons test. Three technical replicates were used for *TERT* mRNA expression

38.5% of *TERTp*$^{WT}$-*IDH*$^{WT}$ GBMs and exhibits characteristics consistent with ALT. The *IDH*$^{WT}$-*TERT*$^{SV}$ GBM subgroup harbors *TERT* structural variants and exhibits increased *TERT* expression. Together, these two subgroups accounted for 82% (32/39) of the *TERTp*$^{WT}$-*IDH*$^{WT}$ GBMs, and exhibited mutual exclusivity ($P = 0.0019$, Fisher's exact test, two-tailed, odds ratio = 0.069). Kaplan–Meier survival analyses revealed that the *IDH*$^{WT}$-ALT (OS: 14.9 months), and *IDH*$^{WT}$-*TERT*$^{SV}$ (OS: 19.7 months) subgroups exhibit poor survival, similar to the *IDH*$^{WT}$-*TERTp*$^{MUT}$ subgroup (OS: 14.74 months). All of these *IDH*$^{WT}$ subgroups displayed shorter OS relative to the *IDH*$^{MUT}$-*TERTp*$^{WT}$ subgroup (OS: 37.08 months, Fig. 3).

**SMARCAL1 mutations contribute to ALT telomere maintenance.** The exome sequencing and ALT results indicate that there is a strong correlation between recurrent somatic

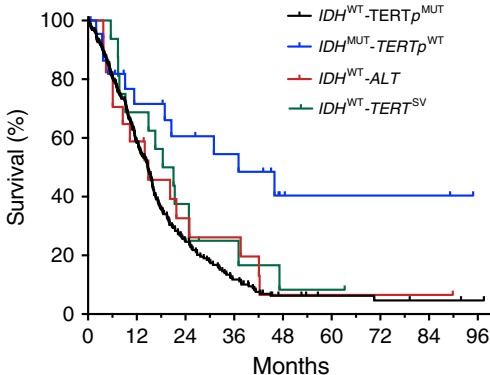

**Fig. 3** New genetic subgroups of GBM display distinct survival patterns. Kaplan-Meier analysis of GBMs grouped by recurrent alterations identified in this study, including *SMARCAL1/ATRX* mutation (*IDH*<sup>WT</sup>-ALT) and *TERT* rearrangement (*IDH*<sup>WT</sup>-*TERT*<sup>SV</sup>). The survival of these new groups are compared to established subgroups of GBM including *TERT* promoter-mutant (*IDH*<sup>WT</sup>-*TERTp*<sup>MUT</sup>, N = 223) and *IDH*-mutant GBMs (*IDH*<sup>MUT</sup>-*TERT*<sup>WT</sup>, N = 23), with median overall survivals of 14.74 and 37.08 months, respectively. Patients in the *IDH*<sup>WT</sup>-other GBM subgroup (N = 7) were excluded due to the limited number of patients. The median OS for the *IDH*<sup>WT</sup>-ALT subgroup (N = 17) was 14.9 months, while the *IDH*<sup>WT</sup>-*TERT*<sup>SV</sup> subgroup (N = 16) had an OS of 19.7 months. Compared to the *IDH*<sup>MUT</sup>-*TERT*<sup>WT</sup> GBMs, the *IDH*<sup>WT</sup>-*TERTp*<sup>MUT</sup> (P = 0.0003, HR = 2.867, 95% CI: 1.929 to 4.262), *IDH*<sup>WT</sup>-ALT (P = 0.0281, HR = 2.302, 95% CI: 1.039 to 5.1), and *IDH*<sup>WT</sup>-*TERT*<sup>SV</sup> GBMs (P = 0.0794, HR = 1.982, 95% CI: 0.8878 to 4.427) have poorer survival. Comparison of survival curves done by log-rank (Mantel–Cox) test

inactivating mutation of *SMARCAL1* and ALT telomere maintenance in a subset of GBMs, similar to the previously established roles of *ATRX* and *DAXX* mutations[13] (Fig. 4a). To further explore the functional connection between somatic *SMARCAL1* mutations and ALT, we identified two cancer cell lines harboring mutations in *SMARCAL1*, D06MG, and CAL-78. D06MG is a primary GBM cell line harboring a nonsense, homozygous *SMARCAL1* mutation (W479X, Supplementary Figure 4D), derived from the tumor of patient DUMC-06. CAL-78 is a chondrosarcoma cell line with homozygous deletion of the first four exons of *SMARCAL1*, resulting in loss of expression (Supplementary Figure 4A–C)[32]. Both *SMARCAL1*-mutant cell lines exhibited total loss of SMARCAL1 protein expression by western blot, with intact expression of ATRX and DAXX (Fig. 4c) and hallmarks consistent with ALT, including ALT-associated pro-myelocytic leukemia (PML) bodies (APBs), DNA C-circles, and ultrabright telomere DNA foci[13,17,33] (Fig. 4b). Restoration of *SMARCAL1* expression in these cell lines significantly reduced colony forming ability, supporting the role of *SMARCAL1* as a tumor suppressor (Fig. 4d, Supplementary Figure 5A–C).

We then investigated the extent to which expression of wildtype (WT) *SMARCAL1* or cancer-associated *SMARCAL1* variants modulate ALT hallmarks in cell lines with native *SMARCAL1* mutations. We found that SMARCAL1 WT expression markedly suppressed ultrabright telomeric foci in both CAL-78 and D06MG. (Fig. 4e). Next, we sought to investigate the effects of somatic *SMARCAL1* variants on C-circle abundance. Cancer-associated mutations tested from our GBM cohort included SMARCAL1 Arg645Ser (R645S), Phe793del (del793), and Gly945fs*1 (945 fs). In addition, we examined mutation patterns in pan-cancer TCGA (The Cancer Genome Atlas) data on cBioportal[34] and found that *SMARCAL1* mutations and homozygous deletions are present at low frequency in several other cancer types (Supplementary Figure 6A). We tested two

SMARCAL1 recurrent variants, R23C and R645C, that were identified from these sequencing studies. R23 (n = 5 mutations) is located in the RPA-binding domain, while R645 (n = 3 mutations) is located in the SNF2 helicase domain, similar to the R645S variant identified in our cohort (Supplementary Figure 6B).

SMARCAL1 WT expression in both CAL-78 and D06MG significantly suppressed C-circle abundance relative to the control condition. In contrast, expression of SMARCAL1 R764Q, a well-studied helicase loss-of-function mutation found in a patient with Schimke immune-osseous dysplasia (SIOD)[35], failed to fully suppress C-circles in CAL-78 and D06MG, demonstrating that SMARCAL1 helicase activity is critical for suppression of these ALT features. Rescue with SMARCAL1 R645S, R645C, and del793 failed to fully suppress C-circles in both cell lines, similar to R764Q. However, overexpression of the SMARCAL1 R23C and fs945 constructs resulted in a similar suppression of C-circle levels to that of the wildtype rescue (Fig. 4g). Notably, the GBM case with SMARCAL1 fs945 mutation from our study exhibited concurrent loss of ATRX expression by IHC, indicating that perhaps ATRX loss was the primary genetic lesion associated with ALT in this case.

Finally, we investigated if knockout of *SMARCAL1* is sufficient to induce hallmarks of ALT in GBM cell lines. We used CRISPR/Cas9 gene editing to generate *SMARCAL1* knockout clones in the ALT-negative GBM cell lines U87MG and U251MG[36,37]. In total, 12 U251MG (A: 5 clones, B: 7 clones) and 10 U87MG (A: 2 clones, B: 9 clones) lines were validated as *SMARCAL1* knockout clones using this approach (Fig. 5a, Supplementary Figure 7A,B, Supplementary Data 11–12). Isogenic *SMARCAL1*<sup>−/−</sup> GBM cell lines were assessed for accumulation of C-circles by dot blot. In both cell lines, 30% of isogenic *SMARCAL1*<sup>−/−</sup> clones isolated exhibited significantly increased levels of C-circles (Fig. 5b), as well as rare ultrabright telomere foci and APBs (Fig. 5c), indicating that loss of *SMARCAL1* in GBM cells can induce signs of ALT.

## Discussion

Approximately one in every five adult GBM patients have tumors that are wildtype for *TERTp* and *IDH1/2*[3,4]. *TERTp*<sup>WT</sup>-*IDH*<sup>WT</sup> GBMs are a poorly understood subgroup that have been defined by an absence of common biomarkers (mutations in *TERTp*, *IDH1/2*, and 1p/19q codeletion). Here, we used genomic sequencing (WES, WGS) and characterization of telomere maintenance mechanisms to define the genetic landscape of *TERTp*<sup>WT</sup>-*IDH*<sup>WT</sup> GBMs and uncover novel alterations associated with telomere maintenance in GBM.

We identified an ALT-positive subgroup of *TERTp*<sup>WT</sup>-*IDH*<sup>WT</sup> GBMs, known as *IDH*<sup>WT</sup>-ALT, which is made up equally of GBMs mutated in *ATRX* (notably without *IDH* or *TP53* mutations) or *SMARCAL1*. Our study reveals a novel role for somatic recurrent loss-of-function alterations in *SMARCAL1* in cancers with the ALT telomere maintenance mechanism. Another recent study[26] reported a role for SMARCAL1 in regulating ALT activity in *ATRX*-deficient cell lines by resolving replication stress and telomere stability[38]. Here, we show that cancers with somatic mutation of *SMARCAL1* are ALT positive, and this represents, to our knowledge, the only other reported gene mutation associated with ALT other than *ATRX* and *DAXX* mutations[13]. Future studies should investigate if ATRX plays a role in the absence of *SMARCAL1* expression at the telomeres in these tumors.

Our results demonstrate the importance of intact SMARCAL1 helicase domains in suppressing characteristics of ALT in *SMARCAL1* mutant, ALT-positive cancer cell lines (Fig. 4g). These findings are consistent with a previous study[27], which used

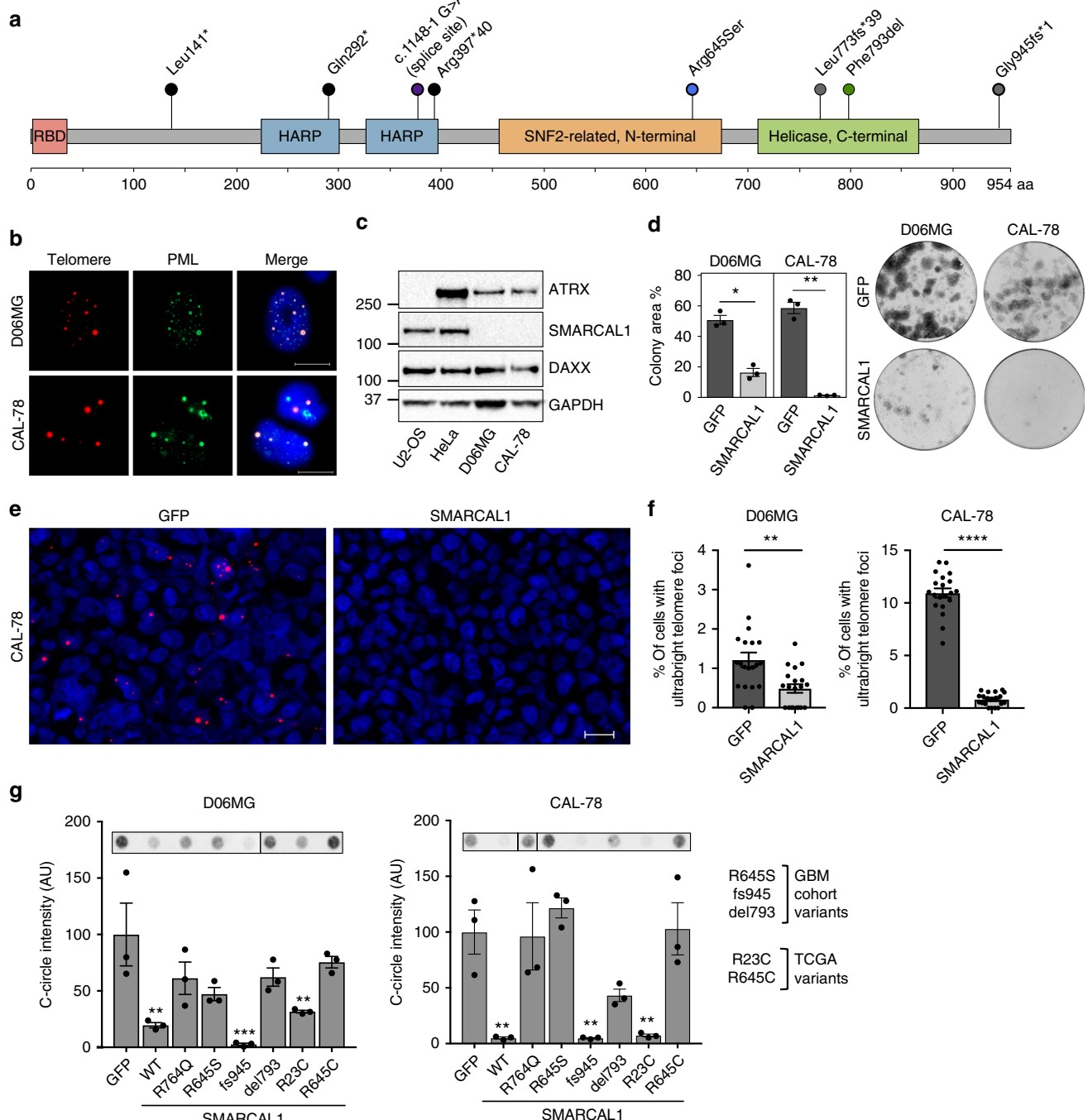

**Fig. 4** Inactivating mutations in *SMARCAL1* mutations cause hallmarks of ALT. **a** The majority of mutations identified in *SMARCAL1* in an expanded cohort (*N* = 39) of *TERTp*^WT-*IDH*^WT GBMs are likely inactivating (frameshift, nonsense). Protein domains of SMARCAL1 are shown (RBD RPA-binding domain, HARP HepA-related protein). **b** We identified two cancer cell lines harboring inactivating mutations in *SMARCAL1*: D06MG (patient-derived GBM, W479X) and CAL-78 (chondrosarcoma, deletion of exons 1–4). These cell lines exhibit signs of ALT, including ALT-associated PML bodies (APBs), as indicated by the co-localization of PML (immunofluorescence) and ultrabright telomere foci (FISH), and the accumulation of C-circles. **c** Western blot confirms the absence of *SMARCAL1* expression in both CAL-78 and D06MG, as well as intact expression of *ATRX* and *DAXX*. Controls include U2-OS (*ATRX*-negative) and HeLa (positive control). **d** Overexpression of *SMARCAL1* significantly decreased (D06MG, *P* < 0.05; CAL-78, *P* < 0.005) colony-forming ability as measured by percent area. **e**, **f** Overexpression of SMARCAL1 dramatically reduces the appearance of ALT-associated ultrabright telomere foci relative to the GFP control (CAL-78 is shown). **g** SMARCAL1 constructs harboring either wildtype, helicase dead (R764Q, from SIOD), mutations from the expanded cohort (R645S, del793, fs945) and recurrent mutations seen in pan-cancer data (R23C, R645C) were assayed for effects on ALT-associated C-circles. The SMARCAL1 helicase domain function is critical for suppression of C-circles, as constructs with mutations in these domains fail to fully suppress markers of ALT, compared to wildtype constructs or SMARCAL1 with mutations in the RPA-binding domain (R23C) or the 945 fs variant. Error bars in **d**, **f**, **g** denote s.e.m. *P* < 0.05; **P* < 0.01; ***P* < 0.001; ****P* < 0.0001; Paired *t*-test (**d**, **f**) and one-way ANOVA with Dunnett's multiple comparisons test (**g**). Scale bar indicates 20 µm. Colony formation and C-circle experiments were performed in triplicate

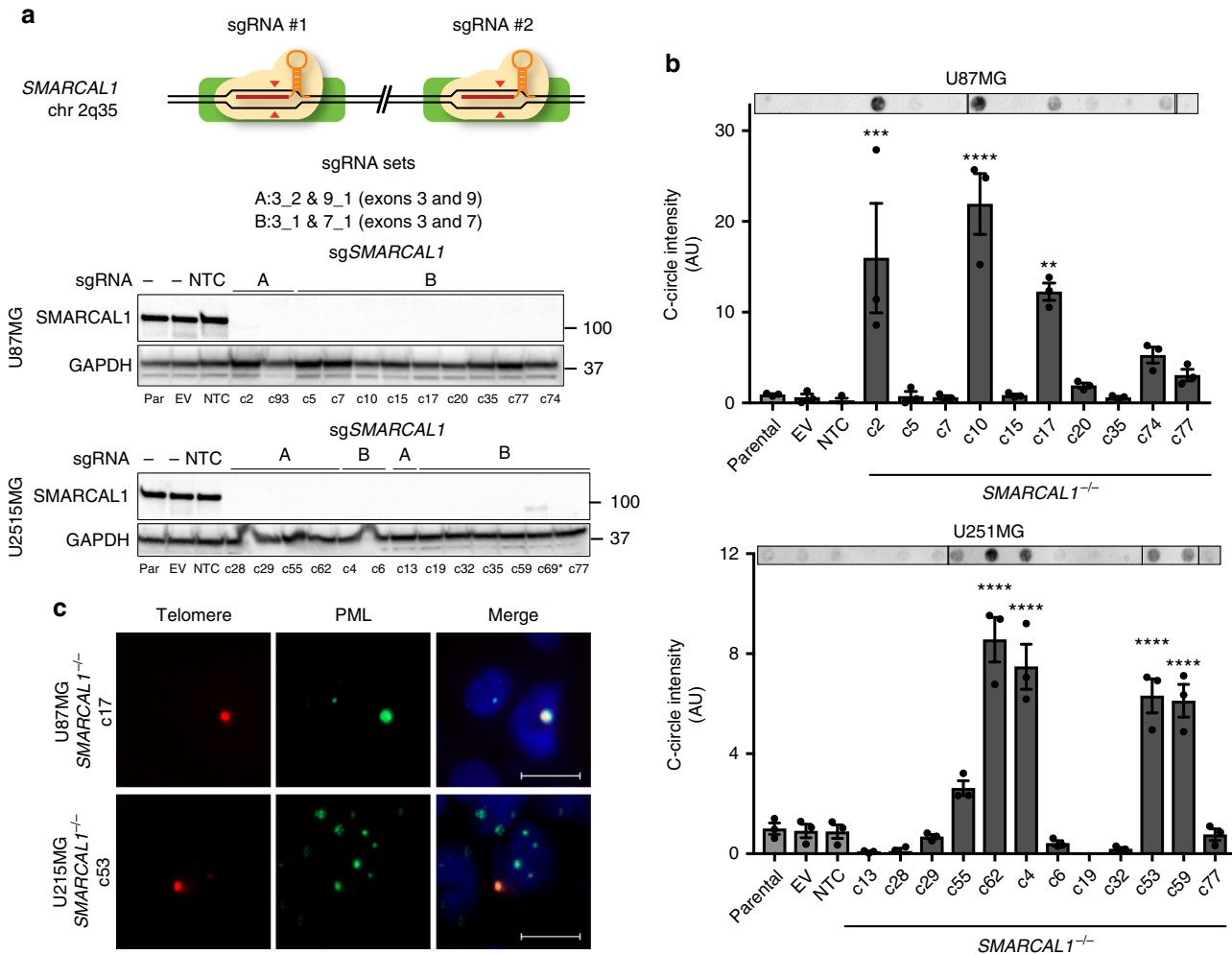

**Fig. 5** Loss of *SMARCAL1* in glioblastoma cell lines leads to features of ALT. **a** CRISPR/Cas9 gene editing was used to generate *SMARCAL1* knockout GBM lines (U87MG and U251MG). Two guide combinations (A: 3_2 & 9_1 and B: 3_1 & 7_1) were used targeting exons 3 and 9 and 3 and 7, respectively. Clones were sequenced and validated as isogenic knockout lines by western blot (*clone c69 was excluded due to faint band). **b** Cell lines were assessed for C-circle accumulation (by dot blot), a characteristic observed in cells using ALT for telomere maintenance. Approximately 30% of isogenic *SMARCAL1* knockout GBM lines isolated in both U87MG and U251MG exhibited significantly increased levels of C-circles (U87MG: 4/12, U251MG: 3/10), as compared to the parental cell line. **c** C-circle-positive *SMARCAL1* knockout clones were assessed for the presence of ALT-associated PML bodies (APBs), as indicated by the co-localization of PML (immunofluorescence) and ultrabright telomere foci (FISH). Rare cells were identified in these C-circle-positive clones with APBs. Error bars in **b** denote s.e.m. **$P < 0.01$; ***$P < 0.001$; ****$P < 0.0001$; one-way ANOVA with Dunnett's multiple comparisons test relative to parental cell line. Scale bar indicates 10 μm. C-circle experiments were performed in triplicate

RNA interference-mediated *SMARCAL1* knockdown in Hela1.3 and *SMARCAL1* gene knockout in MEFs (ALT-negative cell lines with native *SMARCAL1* expression) to investigate the effect of *SMARCAL1* depletion on C-circle abundance. The investigators reported that SMARCAL1-mediated C-circle suppression requires intact helicase activity, and that deletion of the RPA binding domain does not affect C-circle suppression in these cell lines[27].

*SMARCAL1* is recruited to sites of DNA damage and stalled replication forks by RPA, where it promotes fork repair and restart, thereby helping to maintain genome stability[24,25,39,40]. Previous work has shown that bi-allelic germ-line mutations of *SMARCAL1* cause the autosomal-recessive disease SIOD, a rare developmental disorder characterized by skeletal dysplasia, renal failure, T-cell deficiency, and often microcephaly[41]. There is some evidence that SIOD patients have increased risk for cancer[42,43], neurologic abnormalities[44], and chromosomal instability[45]. In the context of our findings, linking *SMARCAL1* alterations to the pathogenesis of ALT-positive

tumors provides insights that may inform the design of therapeutics to exploit the altered replication stress response present in ALT-positive tumors. Additionally, our exome sequencing data show that *SMARCAL1*-mutant GBMs often have mutations in *PTEN*, *NF1*, and *TP53*, which may be necessary co-occurring alterations necessary for gliomagenesis. Our analysis of previous sequencing studies reveals that among diffuse gliomas, *SMARCAL1* mutations appear to be absent in lower-grade gliomas (WHO grade II–III) and only present in GBMs. Furthermore, *SMARCAL1* mutation is not present in the other major genetic subtypes of GBM ($IDH^{MUT}$-$TERTp^{WT}$ or $IDH^{WT}$-$TERTp^{MUT}$)[12,46,47]. *SMARCAL1* somatic mutations occur in other cancer types (Supplementary Figure 6), many of which are known to exhibit ALT in a subset of tumors[17]. We found the mutational pattern in a recent study of sarcoma of particular interest, as this tumor type commonly exhibits ALT. We identified a number of likely pathogenic alterations in *SMARCAL1* in 4% of all cases, including helicase domain mutations with co-existing shallow copy number deletion, as well as tumors with

homozygous deletions (Supplementary Figure 8)[48–50]. Additionally, the *SMARCAL1*-mutated ALT-positive cell line we identified in our study, CAL78, is a chondrosarcoma cell line.

We also identified recurrent *TERT* rearrangements in approximately half of *TERTp*^WT-*IDH*^WT GBMs, now defined as *IDH*^WT-*TERT*^SV GBMs. Recent studies have revealed the presence of similar structural rearrangements upstream of *TERT* in kidney cancer[51] and neuroblastoma[52,53]. As the exact location of the break point was variable (similar to patterns seen in other cancers[51–53]), these alterations may translocate *TERT* to areas of the genome with a genetic environment more permissive to increased *TERT* expression.

Taken together, we have delineated two new genetically defined GBM subgroups, *IDH*^WT-*TERT*^SV and *IDH*^WT-ALT. Similar to the established *IDH*^MUT and *TERTp*^MUT genetic subgroups of GBM[4–8,10], the *IDH*^WT-ALT and *IDH*^WT-*TERT*^SV genetic subgroups exhibit recurrent and distinct genetic alterations leading to either ALT-mediated or telomerase-mediated mechanisms of telomere maintenance (Supplementary Figure 9).

We also observed truncating mutations in the putative oncogene *PPM1D*, similar to previous observations of *PPM1D* mutations in brainstem gliomas[11], suggesting that *PPM1D* is a candidate driver gene in a subset of *TERTp*^WT-*IDH*^WT GBMs. In the TCGA LGG and GBM studies, *PPM1D* truncating mutations were rare (<1% of cases); however, gain or amplification occurred in 5.7% and 12.5% of cases, respectively[23,34,46]. *PPM1D* alterations therefore appear to be present both in brainstem gliomas and less frequently in supratentorial gliomas.

Finally, we identify clinically actionable alterations through sequencing in this cohort, including *BRAF* V600E mutations. While *BRAF* is frequently altered in pediatric gliomas, it is uncommon in adult gliomas (0.7–2%)[46,47,54]. In our study, we identified recurrent BRAF V600E alterations primarily in adult *TERTp*^WT-*IDH*^WT GBM patients 30 years old or younger. These results suggest that *BRAF* mutations may be suspected in young adult *TERTp*^WT-*IDH*^WT GBM patients, which provides an opportunity to use molecular diagnostic markers and targeted BRAF V600E/MEK blockade, which has shown promise in pre-clinical models of astrocytoma[55,56] and in pediatric and adult patients with *BRAF*-mutant tumors[57].

In conclusion, these studies identify novel biomarkers that can be used to objectively define *TERTp*^WT-*IDH*^WT GBM tumors and have discovered a novel role of somatic *SMARCAL1* loss-of-function mutations in the ALT phenotype in human cancers.

## Methods

**Sample preparation and consent**. All patient tissue and associated clinical information were obtained with consent and approval from the Institutional Review Board from The Preston Robert Tisch Brain Tumor Center BioRepository (accredited by the College of American Pathologists). Adult GBM tissues were defined as WHO grade IV gliomas diagnosed after 18 years of age. Tissue sections were reviewed by board-certified neuropathologists to confirm histopathological diagnosis, in accordance with WHO guidelines, and select samples with ≥70% tumor cellularity by hematoxylin and eosin (H&E) staining for subsequent genomic analyses. A total of 25 GBMs were used for WES, and 9 for WGS. Two cases included in this study have previously been sequenced by WES[12], and Sanger sequencing for *TERT* promoter and *IDH1/2* mutational status for 240 GBMs was used to identify candidate *TERT/IDH* wildtype tumors[4]. Patient diffuse glioma tumor samples from Duke University Hospital used in this study were diagnosed between 1984 and 2016.

**DNA and RNA extraction**. DNA and RNA were extracted from homogenized snap-frozen tumor tissue using the QIAamp DNA Mini Kit (QIAGEN) and RNeasy Plus Universal Mini Kit (QIAGEN) per manufacturer's protocols.

**Quantitative RT-PCR**. Reverse transcription was performed using 1–5 μg of total RNA and the RNA to complementary DNA (cDNA) EcoDry Premix (Clontech). RT-PCR for *TERT* expression was performed on generated cDNA in triplicate using the KAPA SYBR FAST (Kapa Biosystems) reagent and the CFX96 (Bio-Rad) for

thermal cycling and signal acquisition. The ΔΔCt method (CFX Manager) was used to determine normalized expression relative to *GAPDH* expression. Primers and protocols are listed in the supplementary material (Supplementary Data 13–14).

**Whole exome sequencing**. Sample library construction, exome capture, next-generation sequencing, and bioinformatic analyses of tumors and normal samples were performed at Personal Genome Diagnostics (PGDX, Baltimore, MD) as previously described[58]. In brief, genomic DNA from tumor and normal samples was fragmented, followed by end-repair, A-tailing, adapter ligation, and polymerase chain reaction (PCR). Exonic regions were captured in solution using the Agilent SureSelect approach according to the manufacturer's instructions (Agilent, Santa Clara, CA). Paired-end sequencing, resulting in 100 bases from each end of the fragments, was performed using the HiSeq2500 next-generation sequencing instrument (Illumina, San Diego, CA). Primary processing of sequence data for both tumor and normal samples was performed using Illumina CASAVA software (v1.8). Candidate somatic mutations, consisting of point mutations, small insertions, and deletions, were identified using VariantDx across the regions of interest. VariantDx examined sequence alignments of tumor samples against a matched normal while applying filters to exclude alignment and sequencing artifacts. Specifically, an alignment filter was applied to exclude quality failed reads, unpaired reads, and poorly mapped reads in the tumor. A base quality filter was applied to limit inclusion of bases with a reported phred quality score of >30 for the tumor and >20 for the normal samples. A mutation in the tumor was identified as a candidate somatic mutation only when: (i) distinct paired reads contained the mutation in the tumor; (ii) the number of distinct paired reads containing a particular mutation in the tumor was at least 10% of the total distinct read pairs; (iii) the mismatched base was not present in >1% of the reads in the matched normal sample; and (iv) the position was covered by sequence reads in both the tumor and normal DNA (if available). Mutations arising from misplaced genome alignments, including paralogous sequences, were identified and excluded by searching the reference genome. Candidate somatic mutations were further filtered based on gene annotation to identify those occurring in protein coding regions. Finally, mutations were filtered to exclude intronic and silent changes, while mutations resulting in missense mutations, nonsense mutations, frameshifts, or splice site alterations were retained. Amplification analyses were performed using a Digital Karyotyping approach through comparison of the number of reads mapping to a particular gene compared to the average number of reads mapping to each gene in the panel. IntOgen analysis was used to identify candidate driver genes. DUMC-14 was excluded from this initially as it had high levels of mutations relative to the rest of the cohort. Candidate drivers were included if they were recurrently mutated ($n \geq 2$, separate cases) and $P < 0.05$ (by OncodriveFM or OncodriveCLUST). Alignments were done to hg18.

**Whole genome sequencing**. The quality of DNA for WGS was assessed using the Nanophotometer and Qubit 2.0. Per sample, 1 μg of DNA was used as input for library preparation using the Truseq Nano DNA HT Sample Prep kit (Illumina) following the manufacturer's instructions. Briefly, DNA was fragmented by sonication to a size of 350 bp, and then DNA fragments were endpolished, A-tailed, and ligated with the full-length adapter for Illumina sequencing with further PCR amplification. PCR products were purified (AMPure XP) and libraries were analyzed for size distribution by the 2100 Bioanalyzer (Agilent) and quantified by real-time PCR. Clustering of the index-coded samples was performed on a cBot Cluster Generation System using the HiSeq X HD PE Cluster Kit (Illumina), per manufacturer's instructions. Libraries were then sequenced on the HiSeq X Ten and 150 bp paired-end reads were generated. Quality control was performed on raw sequencing data. Read pairs were discarded if: either read contained adapter contamination, more than 10% of bases were uncertain in either read, or the proportion of low-quality bases was over 50% in either read. Burrows–Wheeler Aligner[59] (BWA) was used to map the paired-end clean reads to the human reference genome (hg19). After sorting with samtools and marking duplicates with Picard, the resulting reads were stored as BAM files. Somatic single-nucleotide variants were detected using muTect[60] and somatic InDels were detected using Strelka[61]. Copy number variations were identified using control-FREEC[62]. Genomic rearrangements were identified using Delly[30] (v0.7.2). ANNOVAR[63] was used to annotate variants identified.

**Break-apart FISH for *TERT* rearrangements**. Matched formalin-fixed, paraffin-embedded (FFPE) slides were received with one set H&E stained. The tumor location was identified and marked on the slide so that tumor-specific regions could be analyzed. The unstained slides were then aligned with the H&E-stained slides so that potential rearrangements in the tumor zone could be analyzed. Break-apart probes were designed to span *TERT*, with BAC clones mapped (hg19) to chr5: 816,815–1,195,694 (green) and chr5: 1,352,987–1,783,578 (orange) and directly labeled. The break-apart probe set was manufactured with the above design and was first tested on human male metaphase spreads. The probe and the sample were denatured together at 72 °C for 2 min followed by hybridization at 37 °C for 16 h. Slides were then washed at 73 °C for 2 min in 0.4× SSC/0.3% IGEPAL followed by a 2-min wash at 25 °C for 2 min in 2× SSC/0.1% IGEPAL. Slides were briefly air-dried in dark, applied DAPI-II, and visualized under fluorescence

microscope. For FFPE tissue sections, the following pretreatment procedure was used. The sections were first aged for 30 min at 95 °C, deparaffinized in Xylene, dehydrated in 100% ethanol, and air-dried. The slides with the sections were then incubated at 80 °C for 1 h and then treated with 2 mg/ml pepsin in 0.01 N HCl for 45 min. Slides were then briefly rinsed with 2× SSC, passed through ethanol series for dehydration, dried, and used for hybridization. The probe and the sample were denatured together at 83 °C for 5 min followed by hybridization at 37 °C for 16 h. Slides were then washed at 73 °C for 2 min in 0.4× SSC/0.3% IGEPAL followed by a 2-min wash at 25 °C in 2× SSC/0.1% IGEPAL. Slides were briefly air-dried in dark, applied DAPI-II, and visualized under fluorescence microscope. Note that a 5% break-apart signal pattern was arbitrarily considered to be the cut-off for a "Rearrangement" result as the probe is not formally validated on solid tumor tissue at Empire Genomics.

**Cell culture.** CAL-78 was purchased directly from the Deutsche Sammlung von Mikroorganismen und Zellkulturen (DSMZ) and was cultured using RPMI-1640 with 20% fetal bovine serum (FBS). U87, U2-OS and HeLa were purchased from the Duke Cell Culture Facility (CCF), and were cultured with Dulbecco's modified Eagle's medium (DMEM)/F12, McCoy's 5A, and DMEM-HG, respectively, all with 10% FBS. U251MG was a generous gift from the laboratory of A.K.M and was cultured with RPMI-1640 with 10% FBS. D06MG is a primary GBM cell line from resected tumor tissue and was cultured with Improved MEM, Zinc option media, and 10% FBS. All cell lines were cultured with 1% penicillin–streptomycin. Cell lines were authenticated (Duke DNA Analysis facility) using the GenePrint 10 kit (Promega) and fragment analysis on an ABI 3130xl automated capillary DNA sequencer.

**CRISPR/Cas9-mediated SMARCAL1 genetic targeting.** CRISPR guides were designed for minimal off-targets and maximum on-target efficiency for the coding region of SMARCAL1 using the CRISPR MIT[64] (http://crispr.mit.edu) and the Broad Institute sgRNA Design Tools[65] (http://portals.broadinstitute.org/gpp/public/analysis-tools/sgrna-design). Complementary oligonucleotides encoding the guides were annealed and cloned into pSpCas9(BB)-2A-GFP (PX458), which was a gift from Feng Zhang (Addgene plasmid #48138, Supplementary Data 12)[37]. PX458 contains the cDNA encoding Streptococcus pyogenes Cas9 with 2A-EGFP. Negative controls included the parental lines, transfection with empty vector PX458 (no guide cloned), and with PX458-sgNTC[66]. Candidate guides were first tested in HEK293FT by transfecting cloned PX458-sgRNA constructs with lipofectamine 2000 (Life Technologies) according to the manufacturer's guidelines and harvesting DNA from cells 48 h later. These constructs were assessed (i) individually for indel percentage in HEK293FT the Surveyor Mutation Detection Kit (IDT) and (ii) in various combinations for inducing deletions to facilitate gene inactivation and qPCR-based screening for knockout clones (primers and program listed in Supplementary Data 12, S14). Two guides were used to facilitate knockout of SMARCAL1, named sgSMARCAL1 A, which targeted exons 3 and 9 (3_2, 7_1) and B, which targeted exons 3 and 7 (3_1, 7_1). The cell lines U251 and U87 were transfected with Lipofectamine 3000 (Life Technologies) and Viafect (Promega), respectively, and GFP-positive cells were FACS-sorted (Astrios, Beckman Coulter, Duke Flow Cytometry Shared Resource) and diluted to single clones in 96-well plates. Negative control transfected lines (PX458 empty vector and PX458-sgNTC) were not single cell cloned after sorting. Clones were expanded over 2 to 3 weeks and DNA was isolated by the addition of DirectPCR lysis Reagent (Viagen) with proteinase K (Sigma-Aldrich) and incubation of plates at 55 °C for 30 min, followed by 95 °C for 45 min. Then, 1 µl of crude lysate was used as a template for junction-spanning qPCR (to detect dual-sgRNA induced deletion products) with KAPA SYBR FAST (KAPA Biosystems). The junction-spanning amplicon was detected by qPCR signal, using the parental (not transfected) line as a negative control. The targeted exons and junction products were sequenced to validate the presence of indels. Clones were expanded further and screened by western blot to ensure the absence of SMARCAL1 protein expression (Supplementary Figure 7). All relevant programs and primers are listed in Supplementary Data 14–15.

**Lentiviral expression of SMARCAL1.** Lentiviral expression of SMARCAL1 cDNA was done using a constitutive (pLX304) expression vector. pLX304-SMARCAL1 was provided by DNASU (HsCD00445611) and the control pLX304-GFP was a generous gift from Dr. So Young Kim (Duke Functional Genomics Core). Mutagenesis constructs of pLX304-SMARCAL1 (R23C, R645C, R645S, del793, fs945, and R764Q) were generated per the manufacturer's directions using the Quik-Change II Site-Directed Mutagenesis Kit (Agilent). Endotoxin-free plasmids were purified using the ZymoPURE plasmid midiprep kit (Zymo Research) and validated by sequencing and analytical digest. Lentivirus was generated using standard techniques, with the SMARCAL1 cDNA vector, psPAX2 packaging and pMD2.G envelope plasmids in HEK293 and the virus titers were determined using the Resazurin Cell Viability Assay (Duke Functional Genomics Core Facility). Prior to transduction, cell media were replaced with fresh media containing 8 µg/mL polybrene and cells were then spin-infected with lentivirus at a multiplicity of infection of 1 (2250 rpm, 30 min at 37 °C). After 48 h, selection was initiated with blasticidin (pLX304). Transgene expression was confirmed by western blot (Supplementary Figure 6).

**Immunoblotting.** Cells were lysed in protein-denaturing lysis buffer and protein was quantified using the BCA Protein Assay Kit (Pierce). Equal amounts of protein were loaded on SDS-polyacrylamide gels (3–8% Tris-Acetate for blots probing for ATRX, 4–12% bis-tris for all others), transferred to membranes, blocked, and blotted with antibodies. Antibodies used included anti-SMARCAL1 (Cell Signaling Technologies), anti-ATRX (Cell Signaling Technologies), anti-β-Actin (Cell Signaling Technologies), and anti-GAPDH (Santa Cruz Biotechnology) for equal loading control. Original blots are provided in Supplementary Figures 10–11.

**Immunohistochemistry.** Immunolabeling for the ATRX protein was performed on FFPE sections as previously described[67]. Briefly, heat-induced antigen retrieval was performed using citrate buffer (pH 6.0, Vector Laboratories). Endogenous peroxidase was blocked with a dual endogenous enzyme-blocking reagent (Dako). Slides were incubated with the primary antibody rabbit anti-human ATRX (Sigma HPA001906, 1:400 dilution) for 1 h at room temperature and with horseradish peroxidase-labeled secondary antibody (Leica Microsystems), followed by detection with 3,3′-Diaminobenzidine (Sigma-Aldrich) and counterstaining with hematoxylin, rehydration, and mounting. IHC for several cases in the validation cohort was also immunolabeled by HistoWiz Inc. (histowiz.com) using a Bond Rx autostainer (Leica Biosystems) with heat-mediated antigen retrieval using standard protocols. Slides were incubated with the aforementioned ATRX antibody (1:500), and Bond Polymer Refine Detection (Leica Biosystems) was used according to the manufacturer's protocol. Sections were counterstained with hematoxylin, dehydrated, and film coverslipped using a TissueTek-Prisma and Coverslipper (Sakura). Nuclear staining of ATRX was evaluated by a neuropathologist.

**C-circle assay.** C-circle assay was performed as previously described by dot blot[20,68]. Then, C-circles were amplified from 50 ng of DNA by rolling circle amplification for 8 h at 30 °C with φ29 polymerase (NEB), 4 mM dithiothreitol, 1× φ29 buffer, 0.2 mg/mL bovine serum albumin (BSA), 0.1% Tween, and 25 mM of dATP, dGTP, dCTP, and dTTP. C-circles were then blotted onto Hybond-N+ (GE Amersham) nylon membranes with the BioDot (Bio-Rad) and ultraviolet light crosslinked twice at 1200J (Stratagene). Prehybridization and hybridization were done using the TeloTAGGG telomere length assay (Sigma-Aldrich/Roche) and detected using a DIG-labeled telomere probe. DNA from ALT-positive (U2-OS) and -negative (HeLa) cell lines were used as controls.

**Combined immunofluorescence FISH.** Cells were grown on coverslips or µ-slides (Ibidi) to subconfluence and immunofluorescence FISH (IF-FISH) was performed as previously described[69], using the primary antibodies against SMARCAL1 (mouse monoclonal, sc-376377, Santa Cruz Biotechnology, 1:100) and PML (rabbit polyclonal, ab53773, Abcam, 1:200) in blocking solution (1 mg/mL BSA, 3% goat serum, 0.1% Triton X-100, 1 mM EDTA) overnight at 4 °C. Briefly, cells were fixed with 2% formaldehyde. After washing with phosphate-buffered saline (PBS), slides were incubated with goat secondary antibodies against rabbit or mouse IgG, then conjugated with Alexa Fluor 488 or 594 (ThermoFisher, 1:100) in blocking solution. After washing with PBS, cells were fixed again with 2% formaldehyde for 10 min, and washed once again with PBS. Cells underwent a dehydration series (70%, 95%, 100% ethanol), and then incubated with PNA probes (each 1:1000) TelC-Cy3 and Cent-FAM (PNA Bio) in hybridizing solution, denatured at 70 °C for 5 min on a ThermoBrite, then incubated in the dark for 2 h at room temperature. Slides were then washed with 70% formamide 10 mM Tris-HCl, PBS, and then stained with 4′,6-diamidino-2-phenylindole (DAPI) and sealed.

**Telomere FISH.** Deparaffinized slides were hydrated and steamed for 25 min in citrate buffer (Vector Labs), dehydrated, and hybridized with TelC-Cy3 and Cent-FAM (PNA Bio) or CENP-B-AlexaFluor488 in hybridization solution. The remaining steps were done as in combined IF-FISH (above). ALT-positive tumors in FFPE tissue displayed dramatic cell-to-cell telomere length heterogeneity as well as the presence of ultra-bright nuclear foci of telomere FISH signals. Cases were visually assessed and classified as ALT positive if: (i) they displayed ultrabright nuclear foci (telomere FISH signal, 10-fold greater than the signal for individual non-neoplastic cells); and (ii) ≥1% of tumor cells displayed ALT-associated telomeric foci. Areas of necrosis were excluded from analysis. For analysis of ALT status in mutagenesis SMARCAL1 rescue experiments and assessment of ALT status in CRISPR/Cas9 SMARCAL1 knockout experiments, cells were made into formalin-fixed paraffin blocks for easier telomere FISH assessment and quantitative measurement of differences. Briefly, cells were trypsinized, centrifuged onto 2% agarose, fixed in 10% formalin several times to form a fixed cell line plug, then processed, paraffin embedded, and sectioned. For quantitative measurements of differences in ultrabright telomeric foci, telomere FISH-stained slides were scanned at 10× and 20 random fields were selected for assessing the percentage of cells showing ultrabright telomeric foci (~200 cells counted per field).

**1p/19q co-deletion testing.** 1p/19q co-deletion was assessed by either microsatellite-based loss of heterozygosity (LOH) analysis[70] (on DNA extracted from tumor samples and matched germline blood DNA) or by FISH (ARUP labs) on FFPE slides.

**Sanger sequencing**. PCR purification and sequencing reactions were performed by Eton Biosciences or Genewiz using an ABI 3730xl DNA sequencer. PCR reaction conditions and primers are listed in Supplementary Data 14–15.

**Colony-forming assay**. The CAL78-GFP and CAL78-SMARCAL1 cell lines were seeded in triplicate at 2000 cells per well. D06MG-GFP and D06MG-SMARCAL1 cell lines were seeded in triplicate at 1000 cells per well. Cells were fixed with ice-cold methanol and stained with 0.05% crystal violet solution after 15–30 days of incubation. Colony area was quantified using ImageJ and the ColonyArea plugin[71].

**Statistical analysis**. GraphPad Prism 7 and R were used for all statistical analyses (t-test, Kruskal–Wallis test, Fisher's exact test, and Kaplan–Meier curves). Kaplan-Meier analysis was performed for patients with available survival data diagnosed after the year 2000.

**Data availability**. Whole exome sequencing and whole genome sequencing data have been deposited on the Sequencing Read Archive (SRA), accession code: SRP136708.

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

## Acknowledgements

The authors would like to thank Diane Satterfield, Merrie Burnett, and Elizabeth Thomas of The Preston Robert Tisch Brain Tumor Center for assistance in collection of clinical samples. The project was supported by NCI R01CA140316 and NINDS R01NS096407 (PI: Hai Yan), a National Cancer Institute National Research Fellowship Award (1F30CA206423, PI: Bill H. Diplas), and a National Natural Science Foundation Fund (81472559, PI: Yuchen Jiao). Jacqueline Brosnan-Cashman is supported through a postdoctoral fellowship from the Rally Foundation for Childhood Cancer Research and The Truth 365, as well as a National Cancer Institute Training Grant (2T32CA009110-39A1). The authors would like to thank the core facilities used in this study, including the Duke Cancer Institute Flow Cytometry Shared Resource (Lynn Martinek and Michael Cook), the Light Microscopy Core Facility (Yasheng Gao), and the Functional Genomics Core Facility (Sufeng Li and So Young Kim). The authors would like to thank Zachary J. Reitman and Jenna Lewis for their helpful revisions of the manuscript. The authors would like to thank Harini Babu (HistoWiz Inc.) for IHC assistance. Finally, we would like to thank the patients of The Preston Robert Tisch Brain Tumor Center who contributed to this study.

## Author contributions

B.H.D., Y.J., M.S.W., Y.H., A.K.M., and H.Y. designed the study and wrote the manuscript. B.H.D., M.S.W., X.H., J.A.B-C., H.L., Z.W., C.J.M., P.J.K., D.B.L., P.K.G., A.J.R., and R.Y. performed experimental work. B.H.D., M.S.W., X.H., J.A.B-C., L.C., R.M., E.M.L., H.L., S.W., and Y.J. performed data analyses. B.H.D., M.S.W., and H.Y. produced the text and the figures. B.H.D., P.J.K., F.J.R., E.S.L., A.H.F., H.S.F., Y.J., R.E.M, and D.D.B. provided patient materials and data. Y.J., M.S.W., A.K.M., and H.Y. provided leadership for the project. All authors discussed the results and commented on the manuscript.

## Additional information

**Competing interests:** H.Y. is the founder of Genetron Health and receives royalties from Agios and Personal Genome Diagnostics (PGDX). D.D.B. is a scientific advisor for Genetron Health. The remaining authors declare no competing interests.

