## [Peer Review File · Nature Communications]

REVIEWERS' COMMENTS:

Reviewer #1 (Remarks to the Author):

The reviewers have thoughtfully addressed my concerns. I think the manuscript makes an important contribution to our understanding of ALT, cancer, and SMARCAL1.

Reviewer #2 (Remarks to the Author):

The authors have significantly improved the manuscript and addressed my comments. In my opinion the manuscript is now suitable for publication in Nature Communications.

Reviewer #3 (Remarks to the Author):

This is a vastly revised and improved manuscript. The authors addressed all of my concerns and the revision of the title as well as the data provided herein make this work convincing. It is a nice addition to the field that shed light on the pathogenesis of seemingly wild-type GBM. It argues that careful molecular and genetic investigations on large cohorts can help provide the full genomic landscape of these deadly tumors that seem to converge on similar affected pathways through distinct molecular mechanisms.

One minor point, BRAFV600E mutations have been described in young adults with GBM.

We thank the reviewers for their careful reading and comments on our revised manuscript.

Reviewers' Comments:

Reviewer #1 (Remarks to the Author):

The reviewers have thoughtfully addressed my concerns. I think the manuscript makes an important contribution to our understanding of ALT, cancer, and SMARCAL1.

Reviewer #2 (Remarks to the Author):

The authors have significantly improved the manuscript and addressed my comments. In my opinion the manuscript is now suitable for publication in Nature Communications.

Reviewer #3 (Remarks to the Author):

This is a vastly revised and improved manuscript. The authors addressed all of my concerns and the revision of the title as well as the data provided herein make this work convincing. It is a nice addition to the field that shed light on the pathogenesis of seemingly wild-type GBM. It argues that careful molecular and genetic investigations on large cohorts can help provide the full genomic landscape of these deadly tumors that seem to converge on similar affected pathways through distinct molecular mechanisms.

One minor point, BRAFV600E mutations have been described in young adults with GBM.